# Bias-mediated early linearization drives simplicity bias in ReLU networks

## Abstract

ReLU networks and their variants are a key building block of modern deep learning architectures. Despite their ubiquity, our understanding of learning dynamics in these models is still limited. Previous work has relied on a strong set of simplifying assumptions such as the removal of bias terms or predefined gating structures. Here, we explore how the inclusion of bias terms influences learning dynamics in ReLU networks in the rich learning regime. Surprisingly, we observe that the inclusion of bias terms can simplify learning dynamics, i.e. ReLU networks with bias terms have learning dynamics that are strongly aligned to those of well-understood linear models. We explain this similarity by showing that bias terms push ReLU units into a transient linear regime driving subsequent linear learning. We validate the practical scope of our results by showing that bias terms as well as input constants induce linear-like learning on image data and non-linear problems. Importantly, bias-induced early linear learning has substantial implications for simplicity biases: Networks with an early linear phase have a low-rank bias, prefer linear solution on the data, and generalize better. Our results make explicit that bias terms are an important factor that contribute to simplicity biases in neural networks by enhancing preferences for simple solutions. Overall, our findings suggest that observed similarities between linear and nonlinear systems and simplicity biases are connected to the linearizing effect of bias terms.

## 1 Introduction

Behavior and representation in neural networks emerges progressively through the complex interplay of architecture, dataset, and the dynamics of learning. Foundational work in connectionism demonstrated that structure in neural networks emerges in a progressive fashion driven by dataset statistics Rogers & McClelland (2008); Rumelhart et al. (1986). Much theoretical work has explored how *learning dynamics* shape neural network functions and internal representations (Saxe et al., 2014; Jacot et al., 2018; Dominé et al., 2024; Braun et al., 2022) and noted that qualitatively and quantitatively similar representational structure and network functions can emerge in linear and non-linear models (Saxe et al., 2019; Zhang et al., 2025a). Importantly, dynamics in layerwise linear models are nonlinear and the progressive acquisition of dataset structure drives learning dynamics (Saxe et al., 2014; Fukumizu, 1998; Lampinen & Ganguli, 2019). While these simplified models usually do not mimic their nonlinear counterparts exactly, their nonlinear dynamics and theoretical tractability has allowed them to provide explanations for phenomena such as grokking (Kunin et al., 2024), emergence (Nam et al., 2024), specialization of modules (Jarvis et al., 2022; Shi et al., 2022; Sandbrink et al., 2024) or the ability to learn in-context (Zhang et al., 2025b). The study of network dynamics in these simple models has been argued to be a foundational and necessary stepping stone that can ultimately pave the way for analyses of nonlinear models (Nam et al., 2025; Jarvis et al., 2025). A gap remains to understanding the learning dynamics of nonlinear models, however. Theoretical frameworks have been developed which consider ReLU networks as ensembles of linear modules with predefined gating structures (Jarvis et al., 2025; Saxe et al., 2022). Further, Zhang et al. (2025a) established exact agreement in learning dynamics of ReLU and linear models in specific cases, requiring constraints on the data distribution and the removal of bias terms.

A complementary line of work has explored the *simplicity bias* of deep learning. Despite overparameterization, deep neural networks achieve strong generalization performance, challenging the view that such capacity could cause overfitting (Zhang et al., 2021). Importantly, simplicity biases are

also a dynamical phenomenon: Neural networks tend to learn simple functions before complex ones (Hu et al., 2020; Rahaman et al., 2019) and continue to perform well on inputs that are correctly classified by this early simple function even after further training (Kalimeris et al., 2019). These results show that early learning is foundational: it remains intact and shapes the downstream acquisition of the full network function. However, the factors that give rise to this early simplicity and associated downstream consequences have yet to be fully explored. Our work draws a connection between works on (early) simplicity biases in neural networks and qualitative similarities between the learning dynamics of linear and non-linear models by highlighting key contributing factors that can induce linear-like learning as well as simplicity biases in ReLU networks.

This paper examines how the inclusion of bias terms influences learning dynamics ReLU networks and associated downstream consequences. Perhaps surprisingly, we find that the inclusion of bias terms strongly linearizes (early) learning dynamics in ReLU networks. That is, for a fraction of learning, dynamics of ReLU networks are strongly aligned to those of well understood linear models. We also find that this early linear learning has important implications for simplicity biases: ReLU networks with these early linear dynamics prefer linear solutions on the data, Gram matrices of network hidden activations are of lower rank, and activations favor the representation simple task features. We explain this phenomenon by demonstrating that bias terms push ReLU units into a transient linear regime effectively linearizing (early) network dynamics. We also highlight that the phenomenon persists on naturalistic image data and that alignment in these domains can similarly be manipulated via bias terms. **Our contributions are thus as follows:**

- We describe the observation that ReLU and linear networks when equipped with bias terms have equivalent (early) learning dynamics. We highlight that the phenomenon also persists on non-linear tasks and for natural data.

- We explain this phenomenon by showing that early learning of bias terms drives ReLU networks towards a transient regime in which a majority of neurons behave effectively linear. For linear problems, these ReLU networks can be approximately described by analytical solutions for linear networks.

- We show that ReLU with early linear dynamics display enhanced simplicity biases evidenced by lower dimensional representations, an enhanced preference for linear solutions, and the overrepresentation of simple task features.

## 2 RELATED WORK

**Simplicity biases in machine learning.** A broad line of work—spanning theory (Bordelon et al., 2020; Hu et al., 2020) and empirical study (Bhattamishra et al., 2023; Mingard et al., 2023; Shah et al., 2020)—shows that neural networks preferentially discover simple structure. One viewpoint emphasizes the role of the data distribution, suggesting that learning unfolds by progressively capturing dataset moments (Refinetti et al., 2023; Belrose et al., 2024; Huh et al., 2023). A complementary line of work examines optimization dynamics: early in training, networks tend to realize simpler functions (Kalimeris et al., 2019; Refinetti et al., 2023; Rahaman et al., 2019). Further, for certain well-behaved input distributions, nonlinear models without bias terms can behave linear during the very initial stages of learning (Hu et al., 2020). In contrast, our work focuses on architectural and dataset mechanisms that can contribute to the alignment of linear and non-linear models.

**Representational alignment.** The representational alignment between different artificial systems has been extensively explored in the hope of understanding the role of representations in behavior and to enhance model interpretability (Kornblith et al., 2019; Klabunde et al., 2025; Sucholutsky et al., 2024) There is much debate about how much similar patterns of representations relate to computation and behavior (Lampinen et al., 2024; Prince et al., 2025; Lampinen et al., 2025; Braun et al., 2025). Furthermore, the factors that lead to the emergence of similarity have also been debated (Huh et al., 2024). In particular, Lampinen et al. (2024) demonstrated that neural networks represent features relevant to simpler tasks more strongly and that this difference is in part driven by the fact that simpler task features are learned quicker. This observation ties the representational structure of neural networks to the simplicity bias of deep learning. In a simplified setting, we find that bias induced early linear learning enhances the over-representation simple task features.

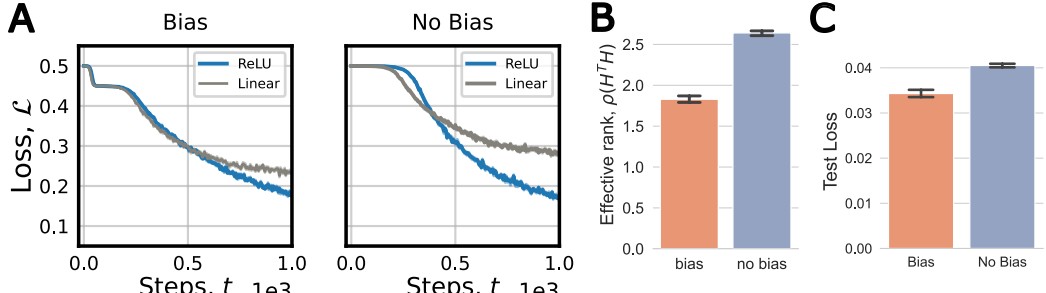

**Figure 1: Transient alignment on MNIST.** We show that bias terms drive alignment on MNIST. **A.** Loss curves of linear and ReLU networks are aligned in early training when models have bias terms (shaded region: SE). **B.** Effective Rank of ReLU networks with and without bias terms. We train networks and terminate training at a loss value of 0.025. At this point, ReLU networks with bias and early linear phase have substantially lower dimensional representations. **C.** Test error. ReLU networks with an early linear phase have lower test error. (5 seeds, error bars 95%-CIs)

## 3 ALIGNMENT ON NATURALISTIC DATA

We begin with some simple experiments on ReLU and linear networks which we train on naturalistic image data. As we will see, this setting provides a useful empirical testbed to understand how and when linear and ReLU dynamics are aligned and how bias terms can drive such similarities.

**Setup.** We train ReLU and linear networks with a single hidden layer on MINST (Li Deng, 2012) and CIFAR-10 (Krizhevsky, 2009) (grayscale) classification tasks. We train networks on a squared error loss using (mini)-batch gradient descent from small initial weights and learning rate. This regime is well known to induce task relevant feature learning. We train models with a single hidden layer of size 512 and we increase the hidden size of the model to 1024 for on CIFAR-10.

As we are interested in factors that drive (early) alignment between ReLU and linear models we consider two different normalizations of our input data. We scale pixel values into the range of $[0, 1]$. For naturalistic data full ablation of bias terms can be challenging as input constants (i.e. constantly active pixels) can effectively act as bias terms (Rubruck et al., 2025). To minimize this effect we subtract each input $\mathbf{x}_i$ by the pixel-wise mean over the full dataset, i.e. $\tilde{\mathbf{x}}_i = \mathbf{x}_i - \bar{\mathbf{x}}$, where $\bar{\mathbf{x}} = \frac{1}{P} \sum_i^N \mathbf{x}_i$. We also manipulate the presence and absence of bias terms on the hidden layer of the model by concatenating inputs with a constantly active pixel $c$ which we set to $c = 10$ (for full implementation details see Appendix E).

We also examine if ReLU networks display a low-rank bias. The measure has been used to characterize simplicity biases and is closely tied to the generalization abilities of neural networks Huh et al. (2023). We compute the effective rank of Gram matrix $\mathbf{H}^T \mathbf{H}$ of hidden representations $\mathbf{H}$ for 500 held out test samples after terminating training at a loss value of 0.025. For a given matrix $\mathbf{A} \in \mathbb{R}^{m \times n}$ the effective rank of a matrix is defined as:

$$\rho(\mathbf{A}) = -\exp\left( \sum_{i=1}^{\min(n,m)} \bar{s}_i log(\bar{s}_i) \right) \tag{1}$$

Here $\bar{s}_i = s_i / \sum_j s_j$ are the normalized singular values of $\mathbf{A}$. The measure is maximized when all singular values are equal and is minimized when a only a single singular value dominates the spectrum.

**Results.** We find that loss curves of ReLU networks with bias terms are closely aligned to those of linear networks in early training. Fig. 1A, left. In contrast, loss curves of diverge quickly when bias terms are ablated Fig. 1A, right. The result demonstrates that bias terms induce an early linear phase in ReLU networks. Corresponding experiments on CIFAR-10 are shown in the Appendix in Fig. 8.

We also find that ReLU networks with early linear learning dynamics display a stronger simplicity bias as quantified by the effective rank $\rho(\mathbf{H}^T \mathbf{H})$. Fig. 1C shows how these networks represent the task in a lower dimensional fashion. Correspondingly the test loss of networks with bias terms

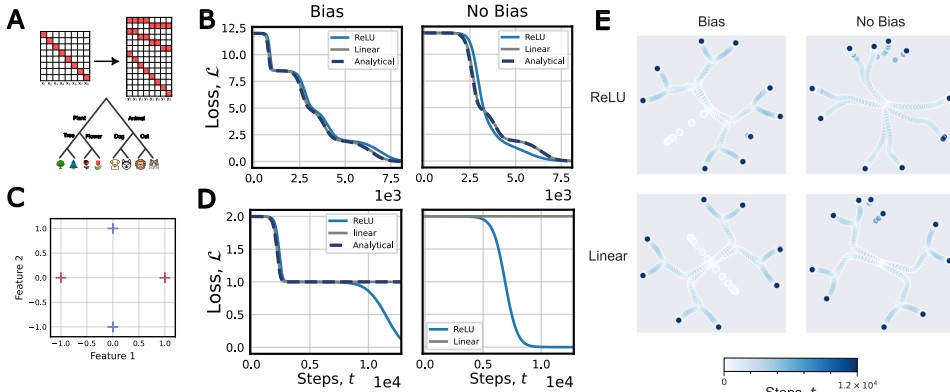

**Figure 2: Alignment on simple tasks. A.** The hierarchical learning task. This problem can be be solved by a linear network. **B.** Loss of ReLU and linear networks. Both network types are functionally similar when equipped with bias terms and display stage-like learning. Also note the good agreement of ReLU networks with exact solutions by (Saxe et al., 2014) devised for linear networks. **C.** A non-linear learning problem. **D.** Networks are also transiently aligned on non-linear problems. **E.** Multi-dimensional scaling of hidden representations. With bias terms, representations in ReLU networks emerge in a structured, linear-like fashion. In non-bias ReLU networks classes separate almost from the beginning of learning.

is lower (Fig. 1C). While previous work (Kalimeris et al., 2019) had conjectured that early learning of simple functions improves model generalization our result causally demonstrate that neural networks with early linear dynamics display a stronger simplicity bias as evidenced by lower-rank representations and improved generalization.

## 4 ALIGNMENT IN SIMPLE SETTINGS.

To isolate this effect, we examine the effect of bias terms on ReLU learning dynamics in a simplified setting.

**Setup.** We first train networks on a linearly solvable semantic learning problem similar to those considered by Saxe et al. (2019). Inputs in the task are encoded as one-hot vectors, such that $\mathbf{X} = \mathbf{I}_P$ and outputs have a hierarchical structure. The problem is visualized in Fig. 2A. Linear networks in this setting display well established stage-like learning. Drops in model loss are driven by the progressive acquisition of SVD modes of the dataset input-output correlation matrix $\tilde{\mathbf{\Sigma}}^{yx}$. For linear networks we then derive analytical solutions according to Saxe et al. (2019) (see Appendix B for a review of these dynamics). We also assess the correspondence on a non-linear problem displayed in Fig. 2C. Full implementation details are described in Appendix E.

**Results.** Surprisingly, we find that ReLU networks with bias terms closely track the dynamics of their linear counterparts, so much so that exact solutions developed for linear networks provide an almost perfect match (although with a slight time shift). However, when removing bias terms the connection breaks down and dynamics between linear and non-linear models diverge. We show the general pattern of our result in Fig. 2B. We also assess the correspondence on a non-linear problem displayed in Fig. 2C. We show that even for non-linear problems bias-ReLU and bias-linear network dynamics are matched in early training (Fig. 2D). As before, the correspondence breaks down when bias terms are ablated. While the focus of this work is on networks trained with bias terms on the hidden layer we show simulations with similar characteristics for biases in both layers in Appendix C.3.

Does the empirically observed functional similarity also translates into a representational alignment of both models? To answer this question, we visualize the evolution of network representations throughout training via Multi-dimensional scaling (Fig. 2E). Hidden representations evolve in an orderly fashion in bias-ReLU networks that show excellent agreement with linear models. The representations for different classes with shared hierarchical features co-evolve, mirroring stage-

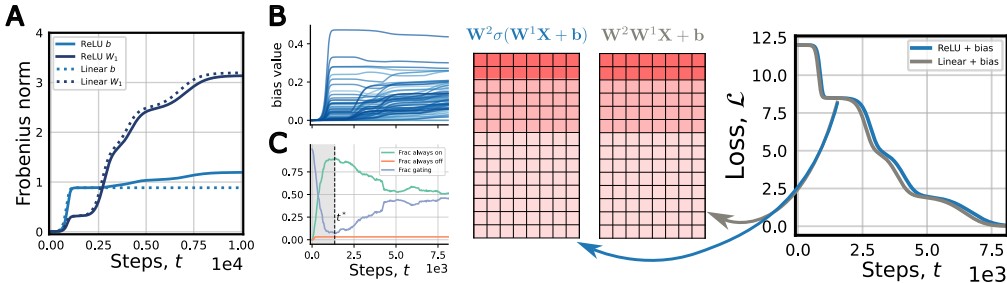

**Figure 3: Mechanisms of bias driven alignment. A.** Frobenius Norm of linear and ReLU network hidden layer weights and biases when training on the task from Fig. 2A. In the beginning of training network biases grow faster than weights and dominate the network function. **B.** Evolution of ReLU network bias terms. Biases are generally positive or around zero and grow fastest early in learning. **C.** Fraction of units that are always-on, always-off, or perform gating. We can see that early in training most units are driven to be always active. With further training, when much variance is already learned, neurons slowly drift back towards the non-linear regime. **D.** The bias-driven network functions of linear and ReLU networks in early training are equivalent to average output statistics $\bar{\mathbf{y}}$.

like learning trajectories. In contrast, representations in bias-free ReLU networks evolve in a less structured manner, with the representations for the different input classes separating almost from the very beginning of learning. The results demonstrate that learning dynamics in bias ReLU networks can give rise to richly structured, linear-like representations. We will now examine the mechanisms that give rise to the observed alignment.

## 5 MECHANISMS OF BIAS-DRIVEN ALIGNMENT

Our main conclusion follows a three-step mechanism: *(i)* when $\bar{\mathbf{y}} := \frac{1}{P}\sum_{i=1}^{P}\mathbf{y}_i \neq \mathbf{0}$, the hidden bias induces a leading "bias mode" that is learned first; *(ii)* this growth pushes most ReLU units into the linear regime; *(iii)* beyond that time learning dynamics are transiently linear. We find that a key difference from a linear net is a small, mode-wise time shift determined by the fraction of dead and non-linear units. Dead neurons are the neurons which are gated-off by the ReLU activation for all data points at initialisation and consequently will remain inactive for the duration of training (Lu et al., 2020).

### 5.1 SETTING

We are studying functional and representational alignment of ReLU and linear networks when these models are equipped with bias terms. We consider a learning task in which a network is presented with input vectors $\mathbf{x}_i \in \mathbb{R}^{N_{in}}$ that are associated to output vectors $\mathbf{y}_i \in \mathbb{R}^{N_{out}}$. The total dataset consists of $\{\mathbf{x}_i, \mathbf{y}_i\}_{i=1}^{P}$ with $P$ samples. We denote $\mathbf{X} = [\mathbf{x}_1, ..., \mathbf{x}_P]$, $\mathbf{Y} = [\mathbf{y}_1, ..., \mathbf{y}_P]$, and $\mathbf{H} = [\mathbf{h}_1, ..., \mathbf{h}_P]$ to denote the full matrices of network inputs, outputs, and hidden activations respectively. We consider two layer linear networks and ReLU networks,

$$\hat{\mathbf{y}}_i^{(\text{lin})} = \mathbf{W}^2(\mathbf{W}^1\mathbf{x}_i + \mathbf{b}), \quad \hat{\mathbf{y}}_i^{(\text{ReLU})} = \mathbf{W}^2\sigma(\tilde{\mathbf{W}}^1\mathbf{x}_i + \mathbf{b}) \tag{2}$$

where $\sigma(x) = \max\{x, 0\}$ denotes the ReLU activation function. Weight matrices are of dimensions $\mathbf{W}^1 \in \mathbb{R}^{N_{hid} \times N_{in}}$, $\mathbf{W}^2 \in \mathbb{R}^{N_{out} \times N_{hid}}$ and the bias vector is of dimension $\mathbf{b} \in \mathbb{R}^{N_{hid}}$. We train our networks to minimize a squared error loss: $\mathcal{L} = \frac{1}{2}\sum_{i=1}^{P}\|\mathbf{y}_i - \hat{\mathbf{y}}_i\|^2$ by full-batch gradient descent with small learning rate $\epsilon$. To incorporate the hidden bias we use *augmented* inputs $\tilde{\mathbf{x}}_i = \begin{bmatrix} c \\ \mathbf{x}_i \end{bmatrix}$. Here $c$ denotes the bias strength which we set to 1 unless specified otherwise. Then the augmented first-layer weights are $\tilde{\mathbf{W}}^1 = \begin{bmatrix} \mathbf{b} & \mathbf{W}^1 \end{bmatrix}$ so that $\hat{\mathbf{y}}_i^{(\text{lin})} = \mathbf{W}^2\tilde{\mathbf{W}}^1\tilde{\mathbf{x}}_i$ and $\hat{\mathbf{y}}_i^{(\text{ReLU})} = \mathbf{W}^2\sigma(\tilde{\mathbf{W}}^1\tilde{\mathbf{x}}_i)$. Letting $\tilde{\mathbf{X}} = [\tilde{\mathbf{x}}_1, ..., \tilde{\mathbf{x}}_P]$ we write the data correlation matrices as

$$\tilde{\mathbf{\Sigma}}^{yx} = \frac{1}{P}\mathbf{Y}\tilde{\mathbf{X}}^T, \qquad \tilde{\mathbf{\Sigma}}^{x} = \frac{1}{P}\tilde{\mathbf{X}}\tilde{\mathbf{X}}^T. \tag{3}$$

We work under the standard simultaneous-diagonalization condition of Saxe et al. (2014), which holds exactly in our one-hot experiments ($\mathbf{X} = \mathbf{I}_P$) and in several other cases we consider: there exist orthogonal $\mathbf{U}, \mathbf{V}$ such that

$$\tilde{\boldsymbol{\Sigma}}^{yx} = \mathbf{U}\,\mathbf{S}\,\mathbf{V}^T, \qquad \tilde{\boldsymbol{\Sigma}}^x = \mathbf{V}\,\mathbf{D}\,\mathbf{V}^T, \tag{4}$$

with diagonal $\mathbf{S} = \mathrm{diag}(s_\alpha)$ and $\mathbf{D} = \mathrm{diag}(d_\alpha)$. For simulations in Figs. 2 to 6 we use a hidden layer size of 64. Given the empirical observation that linear networks and ReLU networks with bias terms show similarities in their *early* learning dynamics, we now turn to provide a theoretical explanation for this effect. We will then illustrate downstream consequences of this bias towards linearity.

## 5.2 THE BIAS MODE DRIVES EARLY LEARNING

We first show the constant input feature (the bias coordinate) couples to the mean label and yields the leading singular mode.

**Proposition 1** (Bias mode). *Given $\tilde{\mathbf{X}}$ as above and suppose $\mathbf{X} = \mathbf{I}_P$ (one-hot inputs). Then $\tilde{\boldsymbol{\Sigma}}^x$ has eigenvalues $\{c^2 + \frac{1}{P}, \frac{1}{P}, \ldots, \frac{1}{P}, 0\}$ where the $\frac{1}{P}$ terms have a multiplicity of $P-1$ and the top eigenvector is $\tilde{\mathbf{v}}_0 \propto [\,1; \frac{1}{cP}\mathbf{1}\,]$. Moreover, if we denote the first singular value of $\tilde{\boldsymbol{\Sigma}}^{yx}$ as $s_0$ then*

$$\tilde{\boldsymbol{\Sigma}}^{yx}\,\frac{\tilde{\mathbf{v}}_0}{\|\tilde{\mathbf{v}}_0\|} \;=\; \sqrt{c^2 + \tfrac{1}{P}}\,\bar{\mathbf{y}}, \qquad \bar{\mathbf{y}} = \frac{1}{P}\sum_{i=1}^{P}\mathbf{y}_i, \qquad s_0 = \sqrt{c^2 + \tfrac{1}{P}}\,\|\bar{\mathbf{y}}\| \tag{5}$$

Full details and proof are in Appendix F.1. Importantly, in deep linear networks the magnitude of singular values of $\boldsymbol{\Sigma}^{yx}$ determine the speed of acquisition for different components of the network function. Specifically, closed-form equations for the training dynamics of the linear network can be derived (Saxe et al., 2014; 2019) by writing the network mapping in terms of the dataset singular vectors ($\mathbf{U}, \mathbf{V}$):

$$\hat{\mathbf{y}}_i^{(\mathrm{lin})} = \mathbf{W}^2(t)\big(\mathbf{W}^1(t)\mathbf{x}_i + \mathbf{b}(t)\big) = \mathbf{W}^2(t)\tilde{\mathbf{W}}^1(t)\mathbf{x}_i = \mathbf{U}\mathbf{A}(t)\mathbf{V}^T\mathbf{x}_i, \tag{6}$$

whose diagonal entries evolve as

$$\mathbf{A}(t)_{\alpha\alpha} = a_\alpha(t) = \frac{s_\alpha/d_\alpha}{1 - \left(1 - \frac{s_\alpha}{d_\alpha a_0}\right)\exp\left(-\frac{2s_\alpha}{\tau}t\right)} \tag{7}$$

The fact that the network mapping can be written in terms of the dataset singular vectors has been labeled the "silent alignment effect" (Atanasov et al., 2021). For a full review of the linear network dynamics see Appendix B. As the *bias mode* is associated with the leading singular value a nonzero $\bar{\mathbf{y}}$ and $c$ implies that this direction of the network function is learned first. Note that this implies that early learning of the bias mode is not dependent on bias magnitude $c$ since it contributes to the largest direction in $\tilde{\boldsymbol{\Sigma}}^{yx}$ for any $c \neq 0$. In Fig. 3D we show how early ReLU and linear network functions output $\bar{\mathbf{y}}$ for all inputs $i$ and in Fig. 10 we highlight that alignment of ReLU networks to linear models occurs even for small values of $c$.

## 5.3 RAPID LINEARIZATION THROUGH BIAS ADAPTATION

Our aim is to understand how early dynamics in ReLU networks are driven by bias terms and the alignment to $\bar{\mathbf{y}}$. Writing $\mathbf{z}_i = \mathbf{W}^1\mathbf{x}_i + \mathbf{b}$, $\mathbf{h}_i = \sigma(\mathbf{z}_i)$ and denoting element-wise multiplication as $\odot$, then full-batch gradient flow gives

$$\dot{\mathbf{W}}^2 = \epsilon\left(\mathbf{Y} - \mathbf{W}^2\mathbf{H}\right)\mathbf{H}^T; \qquad\qquad\qquad \dot{\mathbf{W}}_0^2 \approx \epsilon\mathbf{Y}\mathbf{H}^T \tag{8}$$

$$\dot{\mathbf{W}}^1 = \epsilon\Big(\big[\mathbf{W}^{2T}(\mathbf{Y} - \mathbf{W}^2\mathbf{H})\big] \odot \sigma'(\mathbf{Z})\Big)\mathbf{X}^T; \qquad \dot{\mathbf{W}}_0^1 \approx \epsilon\Big(\mathbf{W}^{2T}\mathbf{Y} \odot \sigma'(\mathbf{Z})\Big)\mathbf{X}^T; \tag{9}$$

$$\dot{\mathbf{b}} = \epsilon\Big(\big[\mathbf{W}^{2T}(\mathbf{Y} - \mathbf{W}^2\mathbf{H})\big] \odot \sigma'(\mathbf{Z})\Big)\mathbf{1}; \qquad \dot{\mathbf{b}}_0 \approx \epsilon\Big(\mathbf{W}^{2T}\mathbf{Y} \odot \sigma'(\mathbf{Z})\Big)\mathbf{1} \tag{10}$$

where $\mathbf{H} = [\mathbf{h}_1, \ldots, \mathbf{h}_P]$ and $\mathbf{Z} = [\mathbf{z}_1, \ldots, \mathbf{z}_P]$. We include the approximation of the gradients at initialization by using the fact that the network output is approximately $\mathbf{0}$ from sufficiently small initial values ($\mathbf{W}_2\mathbf{H} = \mathbf{0}$). The initial effect on $\mathbf{W}_2$ is clear: It will begin to use all active neurons

in the hidden layer to learn the correlation between the initial hidden representations and the output. Given the positivity of ReLU hidden layer, if the outputs are element-wise positive, its weights are driven in the positive direction at initialization. Substituting this small update to $\mathbf{W}^2$ into the updates for $\mathbf{W}^1$ and $\mathbf{b}$, we obtain:

$$\dot{\mathbf{W}}_0^1 \approx \epsilon^2 \Big(\mathbf{H}\mathbf{Y}^T\mathbf{Y} \odot \sigma'(\mathbf{Z})\Big)\mathbf{X}^T; \qquad \dot{\mathbf{b}}_0 \approx \epsilon^2 \Big(\mathbf{H}\mathbf{Y}^T\mathbf{Y} \odot \sigma'(\mathbf{Z})\Big)\mathbf{1} \tag{11}$$

We note that all terms in the bias update equation are element-wise positive semi-definite except for the output correlations $\mathbf{Y}^T\mathbf{Y}$ (which is still symmetric positive semi-definite). Thus, if the outputs are element-wise positive definite (as is common in many tasks, including classification) then the bias term can only grow monotonically and in the case of $\mathbf{X} = \mathbf{I}_P$ for our experiments, so too will $\mathbf{W}^1$. Some additional discussion is presented in Appendix F.2, including what happens when the output contains negative values. The bias $b_j$ and weight $\mathbf{w}_j$ row vector parameters belonging to a node $j$ jointly determine for which input values $\mathbf{x} \in \mathbf{X}$ the neuron is active. We call the boundary between the flat and linear portions of the ReLU the kink. Its distance from the origin in the direction of the weight vector $\mathbf{w_j}$ is given by $k_j = -b_j/||\mathbf{w}_j||$. For a given node $i$, this evolves according to the total differential

$$\dot{k}_j = -\frac{1}{||\mathbf{w}_j||}\dot{b}_j + \sum_k \frac{b_j\,w_{jk}}{||\mathbf{w}_j||^3}\dot{w}_{jk} \tag{12}$$

When the readout weights have positive sign, $\dot{b}_i$ will be positive for positive outputs $\mathbf{Y}$, with the direction of $\dot{w}_{jk}$'s growth depending on interactions between nodes and the input data $\mathbf{X}$. As the norm of the weights vector remains small during early stages of learning (see Fig. 3A), the kink therefore rapidly become strongly negative as the bias increases (see Fig. 12), meaning that for inputs centered around 0 a larger portion of the data enters the linear part (Fig. 3B). Apart from a small subset of neurons that become dead neurons (which occurs when the adaptation of the kink is faster than the adaptation of the sign of the readout weight $w_j^2$) or stay in the nonlinear regime, the majority of neurons therefore rapidly enter a linear regime and become active on all inputs (Fig. 3C).

## 5.4 Time Delays Between ReLU and Linear Network Dynamics

Linear and ReLU networks display a slight offset in their dynamics as seen in in Fig. 2. From Equation 7 we can derive the "hitting time" of a network mode, which is the time taken for the mode to begin learning ($\hat{t}$ such that $a_\alpha(\hat{t}) > \rho$ for a small $\rho \in \mathbb{R}^+$). We derive the full equation for the hitting time in Appendix F.3 and the final equation is:

$$\hat{t} = \frac{\tau}{2s_\alpha}\log\left(\frac{s_\alpha\rho - d_\alpha a_0\rho}{s_\alpha a_0 - d_\alpha a_0\rho}\right) \tag{13}$$

By defining the rate of dead neurons in the hidden layer of ReLU network as $\bar{g}$ and taking the difference in hitting times for the ReLU and linear networks, we are able to obtain the time shift which occurs due to the dead neurons, shown in Equation 14. In Appendix F.3 we prove that this equation is positive definite. Thus, dead neurons always increase the hitting time, explaining the slight time shift between ReLU and linear models observed in Fig. 2.B. It is also important to note that the time delay in Equation 14 treats all neurons that are not dead as if they are effectively linear.

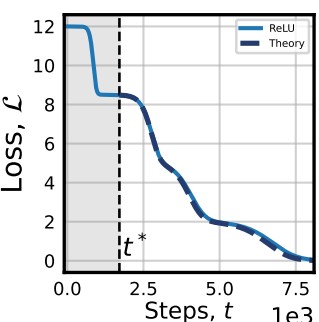

Figure 4: **Relu network dynamics with linear solutions starting from** $t^*$ We see excellent agreement with numerical simulations.

$$\hat{t}_{relu} - \hat{t}_{linear} = \frac{\tau}{2s_\alpha}\log\left(\frac{s_\alpha^2 a_0 - s_\alpha d_\alpha a_0^2\bar{g} - s_\alpha\rho d_\alpha a_0 + d_\alpha^2 a_0^2\bar{g}\rho}{s_\alpha^2 a_0\bar{g} - s_\alpha d_\alpha a_0\bar{g}\rho - s_\alpha d_\alpha a_0^2\bar{g} + d_\alpha^2 a_0^2\bar{g}\rho}\right) \tag{14}$$

Although it is plausible that the time delay should be even greater than predicted when some neurons are only active for a portion of the dataset, in practice the dynamics that are followed on

linear datasets match closely after an initial timeshift. We call $t^\star = \arg\min\{t : f_{\mathrm{on}}(t) = \max_{s \geq 0} f_{\mathrm{on}}(s)\}$ the linearization time, where $f_{\mathrm{on}}(t)$ is the fraction of neurons that are always on, i.e. in the linear regime. Taking this as the starting position, dynamics between ReLU networks and a linear model which incorporates a gating matrix $\mathbf{G}$

$$\hat{\mathbf{y}}_i^{(\mathrm{ReLU})} \approx \mathbf{W}^2(t)\,\mathbf{G}\,\tilde{\mathbf{W}}^1(t)\,\tilde{\mathbf{x}}_i = \mathbf{P}(t)\mathbf{x}_i \tag{15}$$

match (almost) exactly (see Figure 4). Here, we set a fixed gating variable $\mathbf{G} = \mathrm{diag}(\mathbf{g})$ where $g_j = \mathbb{1}[j \in A(t^\star)]$ describes if node $j$ is active on all inputs $i$ at time $t^*$, we then set the initial mode strengths for the deep-linear ODEs using $a_\alpha(t^\star) = \left(\mathbf{U}^T \mathbf{P}(t^\star)\mathbf{V}\right)_{\alpha\alpha}$.

## 6 DOWNSTREAM CONSEQUENCES OF EARLY LINEARITY

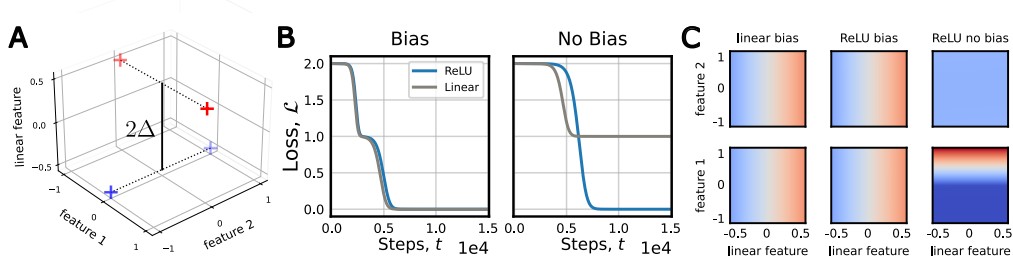

Figure 5: **Preferential linear learning in bias ReLU networks. A.** An dataset which has a non-linearly solvable XOR structure in the first two dimension, the third dimension represents a linear feature with margin $2\Delta = 1$. ReLU networks can solve the task using either linear, non-linear features, or a combination. **B.** Evolution of loss curves when training models with and without bias terms. Loss curves for ReLU and linear networks are closely aligned, when models have bias terms. **C.** Visualization of the learned network functions. We evaluate the networks on all points in the cube and collapse (average) over one dimension. We can see that ReLU and linear networks with bias terms strongly rely on linear features, while ReLU networks with bias terms do not.

### 6.1 PREFERENTIAL LEARNING OF LINEAR SOLUTIONS

ReLU networks are expressive enough to learn non-linear solutions on the data, but in cases where a linear solution exists, they may still settle on a linear solution. We examine a problem considered by Jarvis et al. (2025) that permits both types of solutions. Our aim is to understand if early linearity in ReLU networks biases networks to learn linear solutions. We illustrate the task in Fig. 5A, the first two coordinates form an XoR configuration, while the third introduces linear separability with a margin of $2\Delta$. A linear solution is only possible when $\Delta > 0$. We train ReLU networks with and without bias terms on the task with $\Delta = 0.5$.

Previous work demonstrated, that ReLU networks transition to non-linear solutions well before the problem becomes not linearly solvable (Jarvis et al., 2025). In contrast, our results reveal that bias terms nudge ReLU networks towards linear solutions. We find that loss curves for ReLU networks with bias terms closely resemble linear networks with a small time-shift (Fig. 5B). In contrast, ReLU networks without bias make use of the non-linear structure. We visualize the decision boundaries learned by these models in Fig. 5C. In Appendix C.5 we illustrate the correspondence for a range of values for $\Delta$. We find that ReLU networks with bias terms have an enhanced preference for linear solutions and that dynamics of ReLU and linear models only diverge for relatively small values of $\Delta$. Overall, ReLU networks with bias terms have an enhanced simplicity bias as evidenced in their preference for the acquisition of linear solutions.

## 6.2 SIMPLE BEFORE COMPLEX LEARNING

Learning order contributes to "representational biases" in which simple features explain large amount of variance in representations (Lampinen et al., 2024). To illustrate if early, linear-like learning contributes to such biases we design a task (see Fig. 6A) in which networks solve a linear and non-linear problem in parallel. Fig. 6B shows how bias-ReLU networks are aligned to linear networks in early training and initially only acquire the linear problem. The ReLU loss curve in Fig. 6B, (Bias) settles at the dotted line which indicates the loss value that is achieved when only solving the linear task for a lengthy period. ReLU networks without bias also appear to learn slower after learning the linear task. However, the saddle point is substantially less pronounced.

Analogous to our experiments in Fig. 1C, we examine if bias-ReLU networks display a low-rank simplicity bias. We compute the effective rank of Gram matrix of hidden representations $\mathbf{H}^T\mathbf{H}$ at convergence. We find that bias-ReLU networks have a strong low-rank bias and represent the task in a fundamentally lower dimensional fashion (Fig. 6C). Further interrogation of model representations reveals that early linearity also contributes to the overrepresentation of linear features. We follow Lampinen et al. (2024) and fit linear regressions that predict hidden representations for each of the four input examples from binary inputs which represent each input examples in terms of the linear and nonlinear task. We then assess the variance explained $R^2$ by these regressions. A detailed explanation of the analysis is in Appendix D. Fig. 6D shows that ReLU networks with bias terms encode the linear task more strongly than their bias-free counterparts, despite zero loss on both tasks. However, both models represent linear features more prominently than non-linear features. For bias-free ReLU we also find that less overall representational variance is explained when fitting a regression that contain both regressor, perhaps, hinting at less structured and task-attuned representations.

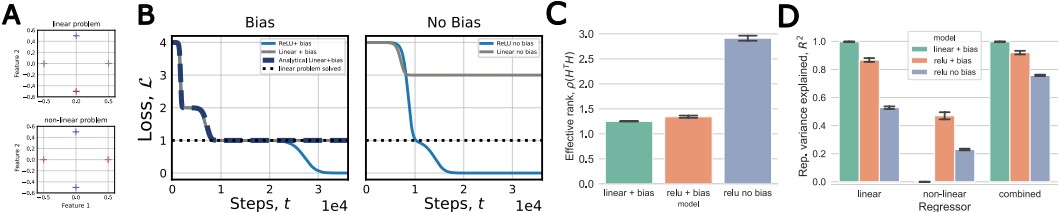

**Figure 6: Linear to non-linear learning in ReLU networks. A.** Models are trained on a linear problem (top) and a non-linear problem (bottom) problem in parallel. Problems are defined on the same four data-points. **B.** With bias (left) ReLU networks are aligned to linear models in early training and learn the linear task (dotted line) first. **C.** Representational variance explained by both tasks. More variance in the representations of bias-ReLU networks is explained by the linear task than for non-bias ReLU networks. While ReLU networks without bias also represent features relevant to the linear task more strongly, the early linear alignment of bias-ReLU networks to linear models boosts the phenomenon. (Averages for 10 seeds, error bars 95%-CIs.)

## 7 CONCLUSION AND FUTURE WORK

In this work, we documented early similarities between ReLU and linear network dynamics when models are equipped with bias terms. We characterized that early linearity enhances simplicity biases in model behaviors and representation. We explain this observation through the early growth of bias terms which pushes a large fraction of ReLU units into a transient linear regime which allows for an approximately linear description of subsequent learning dynamics. Prior analyses of learning dynamics in linear models is usually formulated with the aspirational goal that insights might one day apply to non-linear systems. Our work is taking a step in this direction by showing that, in certain cases, learning dynamics in linear and ReLU networks can, at least temporarily, coincide. ReLU networks are capable of learning non-linear tasks and we document similarity only in the early, linear phase of training. Our analysis is hence restricted and does not provide insights into dynamics that unfold outside of this linear phase. Further, most of our analysis is focused small models trained on relatively simple datasets. However, as rectified units with bias terms are a universal component in almost all modern deep learning architecture we believe that many of our insights plausibly extend to deeper models trained in more practical settings.

**Ethics Statement**  This work is theoretical and uses only small-scale models in controlled experiments. We do not anticipate safety risks from the results.

**Reproducibility statement**  We aim to make our results easy to verify and reproduce. Implementation details (model definitions, training procedures, and hyperparameters) are provided in Appendix E. Descriptions of all datasets are also provided in Appendix E. Proofs are provided in Appendix F. Configuration files, and notebooks to regenerate the main figures are included in the anonymous supplementary materials.

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

## A  APPENDIX

**Overview.**  We structure our appendix as follows: We first provide an overview of exact solutions for deep linear neural networks in Appendix B. We provide additional figures in Appendix C. In Appendix D we explain how we performed the analyses on representational variance explained displayed in Fig. 6D. Appendix E details the Hyperparameters and datasets used in all of our experiments. Finally, Appendix F contains proofs and derivations.

**The use of large language models**  Large language models were used as general assist tools. Models were chiefly used to aid in polishing and editing of the manuscript.

## B  A REVIEW OF EXACT SOLUTIONS IN LINEAR NEURAL NETWORKS

We will quickly review the derivation of exact learning dynamics in deep linear networks by Saxe et al. (2014; 2019). Consider the same setup as outlined in our Appendix 5, **setting**. When training networks using full batch gradient descent in the gradient flow regime dynamics in linear networks dependent on the dataset input-output and input-input correlation matrices. Using singular value decomposition (SVD), these matrices can be expressed as

$$\mathbf{\Sigma}^{yx} = \frac{1}{P}\mathbf{Y}\mathbf{X}^T = \mathbf{U}\mathbf{S}\mathbf{V}^T, \quad \mathbf{\Sigma}^x = \frac{1}{P}\mathbf{X}\mathbf{X}^T = \mathbf{V}\mathbf{D}\mathbf{V}^T. \tag{16}$$

Here $\mathbf{X} \in \mathbb{R}^{N_{in} \times N}$ and $\mathbf{Y} \in \mathbb{R}^{N_{out} \times N}$ contain the full set of input and output vectors. Considering the quadratic loss from the setup section this implies updates of

$$\Delta\mathbf{W}^1 = \epsilon P\mathbf{W}^{2^T}(\mathbf{\Sigma}^{yx} - \mathbf{W}^2\mathbf{W}^1\mathbf{\Sigma}^x); \quad \Delta\mathbf{W}^2 = \epsilon P(\mathbf{\Sigma}^{yx} - \mathbf{W}^2\mathbf{W}^1\mathbf{\Sigma}^x)\mathbf{W}^{1^T}. \tag{17}$$

As we are training with a small learning rate $\epsilon$ we can take a continous time limit of this equation to get the mean change in weights

$$\tau\frac{d}{dt}\mathbf{W}^1 = \mathbf{W}^{2^T}(\mathbf{\Sigma}^{yx} - \mathbf{W}^2\mathbf{W}^1\mathbf{\Sigma}^x); \quad \tau\frac{d}{dt}\mathbf{W}^2 = (\mathbf{\Sigma}^{yx} - \mathbf{W}^2\mathbf{W}^1\mathbf{\Sigma}^x)\mathbf{W}^{1^T}. \tag{18}$$

Here $\tau = \frac{1}{p\epsilon}$ is the learning time constant and $t$ give the units of learning in terms of epochs. Above we have assumed that the right singular vectors $\mathbf{V}^T$ of $\mathbf{\Sigma}^{yx}$ diagonalize $\mathbf{\Sigma}^x$. To decouple dynamics of weights in the network into one dimensional systems we then perform a change of variables utilizing the singular vectors of $\mathbf{\Sigma}^{yx}$ and eigenvectors $\mathbf{\Sigma}^x$.

$$\mathbf{W}^2 = \mathbf{U}\bar{\mathbf{W}}^2\mathbf{R}^T; \quad \mathbf{W}^1 = \mathbf{R}\bar{\mathbf{W}}^1\mathbf{V}^T \tag{19}$$

with $\mathbf{R}$ as an arbitrary orthogonal matrix. After substituting into continous time gradient descent updates Eq. (18) and with some simplification this yields

$$\tau\frac{d}{dt}\bar{\mathbf{W}}^1 = \bar{\mathbf{W}}^{2^T}(\mathbf{S} - \bar{\mathbf{W}}^2\bar{\mathbf{W}}^1\mathbf{D}); \quad \tau\frac{d}{dt}\bar{\mathbf{W}}^2 = (\mathbf{S} - \bar{\mathbf{W}}^2\bar{\mathbf{W}}^1\mathbf{D})\bar{\mathbf{W}}^{1^T}. \tag{20}$$

Here $\mathbf{S}$ and $\mathbf{D}$ are diagonal matrices, if $\bar{\mathbf{W}}^2$ and $\bar{\mathbf{W}}^1$ are also diagonal these dynamics decouple into independent systems. The derivation of Eq. (20) requires the alignment of weight matrices with singular vectors of our correlation matrices from above. However, when we do not initialize our networks with weights that are aligned apriori. However, it has been observed that when training networks from small initial weights that weight singular values will rapidly align to match input correlation singular vectors before network loss changes in any appreciable way (Atanasov et al., 2021). When initializations are balanced, i.e. when the values on the diagonal of $\bar{\mathbf{W}}^1_{\alpha\alpha} = \omega^1_\alpha$ and $\bar{\mathbf{W}}^2_{\alpha\alpha} = \omega^2_\alpha$ are approximately equal. We have $\omega^1_\alpha = \omega^2_\alpha$. When initializing networks from small weights this assumption will hold true in practice. We can then track the general strength of a particular mode of the dataset that is being learned via a scalar $\omega_\alpha = \omega^2_\alpha\omega^1_\alpha$

$$\tau\frac{d}{dt}\omega_\alpha = 2\omega_\alpha(s_\alpha - \omega_\alpha) \tag{21}$$

where $s_\alpha$ is the $\alpha$th singular value on the diagonal of $\mathbf{S}$. Integrating this equation and rearranging terms (full details Saxe et al. (2019)) then gives the evolution of mode strengths. At each time-step $a_\alpha(t)$ then follows a sigmoidal trajectory

$$a_\alpha(t) = \frac{s_\alpha/d_\alpha}{1 - (1 - \frac{s_\alpha}{d_\alpha a_0})e^{-\frac{2s_\alpha}{\tau}t}} \tag{22}$$

here $d_\alpha = \mathbf{D}_{\alpha\alpha}$ are eigenvalues of $\mathbf{\Sigma}^x$, $a_0$ are the singular values at initialization, and $\tau = \frac{1}{N\epsilon}$ is the time constant where $\epsilon$ is the learning rate. Populating a diagonal matrix with the effective singular values $\mathbf{A}(t)_{\alpha\alpha} = a_\alpha(t)$ gives full evolution of network weights for deep linear networks as

$$\mathbf{W}^2(t)\mathbf{W}^1(t) = \mathbf{U}\mathbf{A}(t)\mathbf{V}^T. \tag{23}$$

## C  ADDITIONAL FIGURES.

### C.1  ALIGNMENT ON MNIST WITH STANDARD PREPROCESSING

For completeness, we also consider the more standard approach to preprocessing, i.e. classic standardization. For each input $\mathbf{x}_i$ we subtract the mean and divide by the standard deviation taken over all images and pixels. We then train networks as in Appendix 3, full details in Appendix E.

When using standard preprocessing ( Fig. 1B), bias terms do not play a fundamental role and both types of networks display similar early dynamics. The results demonstrate that bias terms (or input constants) drive the observed alignment between ReLU and linear networks. The observation makes the implicit effect of input constants explicit and highlight how early alignment between ReLU and linear networks cannot be directly manipulated for certain datasets.

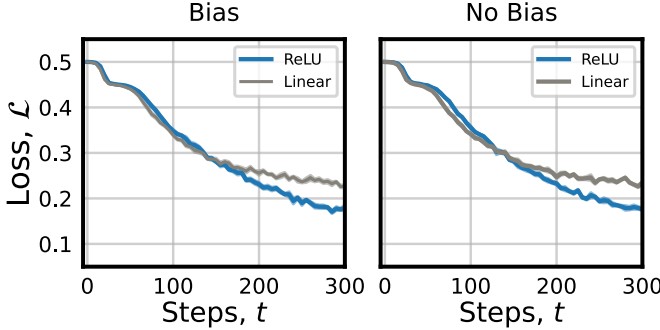

**Figure 7: Transient alignment on MNIST with classic preprocessing.** With standard preprocessing early learning dynamics of linear and ReLU networks align independently of the inclusion of bias terms (shaded region indicates SE).

### C.2  ALIGNMENT ON CIFAR-10

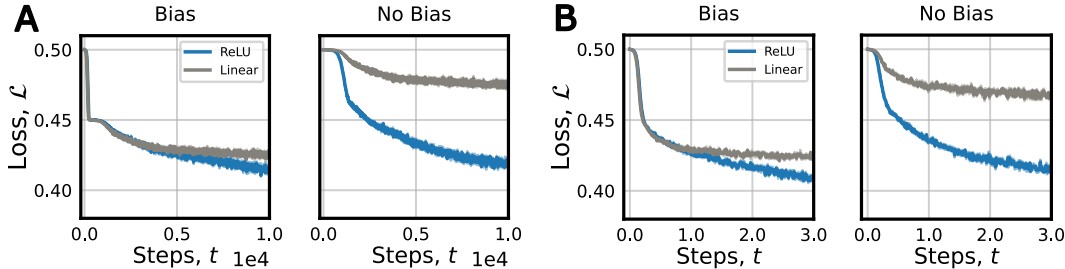

**Figure 8: Transient alignment on CIFAR-10.** We show that bias terms or input constants drive alignment on MNIST. **A.** With pixel-wise mean subtraction, loss curves of linear and linear networks are closely aligned in early training when models have bias terms in the hidden layer. **B.** With standard preprocessing, biases also drive early alignment on CIFAR 10. This is in contrast to Fig. 1B in which input correlations appeared strong enough to drive early alignment alone. (shaded region indicates SE).

### C.3 ALIGNMENT WITH BIAS TERMS IN BOTH LAYERS.

For completeness, we also examine how dynamics are aligned in cases where networks contain bias terms in both layers. I.e. $\hat{\mathbf{y}}_i = \mathbf{W}^2\sigma(\tilde{\mathbf{W}}^1\mathbf{x}_i + \mathbf{b_1}) + \mathbf{b_2}$. We find that dynamics in these cases are also aligned and ReLU networks display characteristic, stage-like learning.

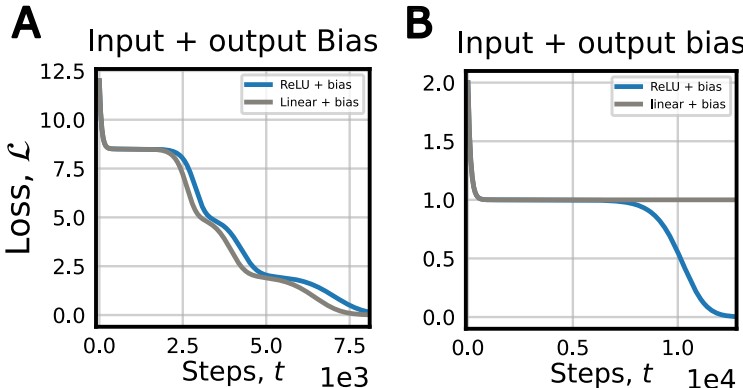

Figure 9: **Alignment of ReLU and Linear networks with bias terms in both layers. A.** Stage-like learning in ReLU networks that have bias terms on both layers on the hierarchical learning task in Fig. 2A. Dynamics of linear and ReLU networks are preserved with the exception of a small time shift. **B.** Early dynamics are also aligned on the non-linear problem from Fig. 2C.

### C.4 THE INFLUENCE OF BIAS MAGNITUDE

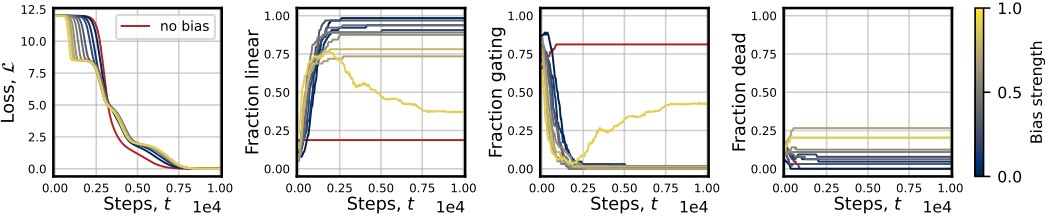

Figure 10: **The influence of bias magnitude on linear-like learning.** Networks are trained on the hierarchical learning task in Fig. 2A with varying degrees of bias strength $c$. **(Left)** We find that loss curves display characteristic stage-like learning in all cases where $c > 0$ **(Left-center)** Fraction of neurons that are effectively linear as a function of $c$. We find that bias terms drive networks towards linearity early in training irrespective of bias magnitude $c$ unless $c = 0$. **(Right-center)** Same as before but plotting the fraction of effectively non-linear units that perform gating on the input. **(Right)** Fraction of effectively dead units.

An intuition on might have is that bias magnitude modulates the degree of alignment to linear learning. Here, we demonstrate that this intuition can be incorrect when training on a linearly solvable problem. We train ReLU networks while varying bias magnitude $c$ across runs on the hierarchical learning task in Fig. 2A. In the main text we considered the alignment of linear and ReLU networks when bias strength equal to a fixed constant $c = 1$. We observe that all networks which have $c > 0$ display characteristic linear aligned, stage-like learning ( Fig. 10, left) and only networks with no bias term significantly diverge from linear dynamics (Fig. 10, left see red line). Further we observe that the fraction of neural network units that perform effectively linear shoots up early in training whenever $c > 0$ (Fig. 10, left-center), conversely the number of neurons which perform gating is small unless $c = 0$ (Fig. 10, right-center). These results illustrate how bias terms drive linear aligned learning in ReLU networks even when bias terms are small.

## C.5 THE INTERACTION OF LINEAR SEPARABILITY AND LINEAR LEARNING

Here, we visualize the alignment of linear and ReLU networks with bias terms on the extended XOR task from Fig. 5A. We highlight that the learning dynamics of both models only diverge for relatively small offsets $\Delta$. Even for very small offsets the dynamics are preserved during the initial bias learning phase.

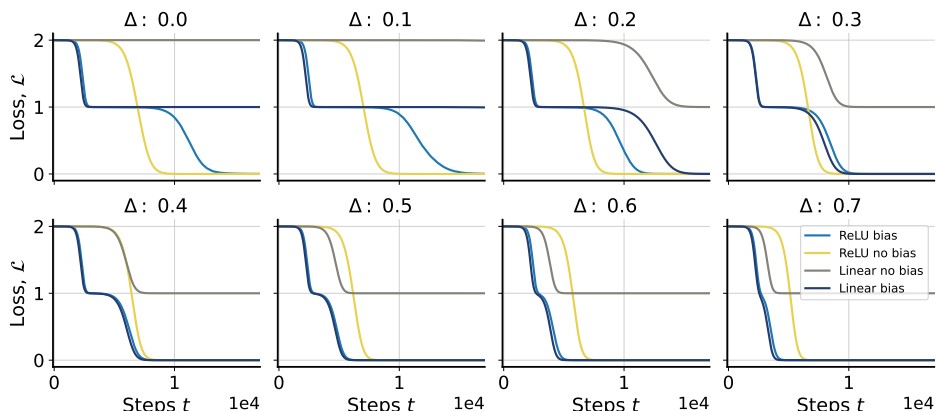

**Figure 11: Visualising preferential linear learning in bias ReLU networks across offsets $\Delta$.** As we increase the linear separability of the two classes in Fig. 5A we find that dynamics of linear and ReLU network with bias terms only diverge for small values of $\Delta$. Overall, bias terms robustly push models towards a preference for linear task solutions.

## C.6 KINK VALUES

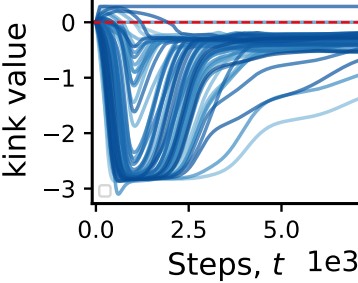

**Figure 12: The evolution of network kink values in ReLU networks.** ReLU networks with bias terms are trained on the task from Fig. 2. The vast majority of kink values rapidly grow negative early in learning. The red dashed line indicates 0.

## D QUANTIFYING REPRESENTATIONAL VARIANCE EXPLAINED

We consider the task displayed in Fig. 6A with $P = 4$ inputs $\mathbf{x}_1, \ldots, \mathbf{x}_P$. Let the hidden representation for input $\mathbf{x}_i$ be $\mathbf{h}_i \in \mathbb{R}^d$, and collect them as $\mathbf{H} = [\mathbf{h}_1, \ldots, \mathbf{h}_P] \in \mathbb{R}^{d \times P}$. We ask how much of the variance across inputs in $\mathbf{H}$ is explained by two label-derived features:

$$\text{simple (linear): } \mathbf{s} = [1, 1, 0, 0]^T, \qquad \text{complex (non-linear): } \mathbf{c} = [1, 0, 0, 1]^T.$$

We fit linear regressions to find the optimal linear predictor $\widehat{\mathbf{W}}$ of the model representations from these features for each of our designs $\mathbf{X} \in \{\mathbf{s}, \mathbf{c}, [\mathbf{s}\,\mathbf{c}]\}$:

$$\widehat{\mathbf{W}} = \min_{\mathbf{W}} \left\| \tilde{\mathbf{Y}} - \tilde{\mathbf{X}}\mathbf{W} \right\|_F^2,$$

with $\mathbf{W} \in \mathbb{R}^{q \times d}$ ($q = 1$ for a single feature, $q = 2$ for the combined case). The proportion of representational variance explained is reported as

$$R^2(\tilde{\mathbf{X}}) \;=\; 1 - \frac{\left\| \tilde{\mathbf{Y}} - \tilde{\mathbf{X}} \widehat{\mathbf{W}} \right\|_F^2}{\left\| \tilde{\mathbf{Y}} \right\|_F^2}.$$

# E    Implementation Details

All networks we train are initialized with small random weights. We sample weights i.i.d. from a normal distribution $\mathcal{N}(0, \sigma^2)$ we specify the initialization scale and other hyper parameters used for each figure below.

**Figure 1.** The networks are trained in MNIST with the preprocessing outlined in Appendix 3. We train on a standard 10-way classification task with one hot labels on a squared error loss. We use the whole standard MNIST training set of 50000 images. We use 500 images from the test set for assessment of test error and computation of gram matrices over hidden representation. Networks are trained with a single hidden layer with hidden size set to 512. The initialization scale $\sigma^2 = 2 \times 10^{-4}$. We train network with (mini) batch SGD with a batch size of 64 and set the learning rate to $5 \times 10^{-4}$. For the assessment of the effective dimensionality of Gram matrices we train networks until the average the sample-wise loss goes to 0.025 at which point we terminate training.

**Appendix Figure 7** The networks are trained in MNIST with the preprocessing outlined in Appendix C.1. We train on a standard 10-way classification task with one hot labels on a squared error loss. We use the whole standard MNIST training set of 50000 images. Networks are trained with a single hidden layer with hidden size set to 512. The initialization scale $\sigma^2 = 2 \times 10^{-4}$. We train network with (mini) batch SGD with a batch size of 64 and set the learning rate to $5 \times 10^{-4}$.

**Figure 2-5. & Appendix Figure 9, 10 & 12.** All networks are trained on the hierarchical task depicted in Fig. 2A. With the exception of loss curves shown in Fig. 2D for which we training on the XOR problem which is depicted in Fig. 2C. For the XOR problem labels are represented in a one-hot fashion. I.e. $[1, 0], [1, 0], [0, 1], [0, 1]$. The initialization scale $\sigma^2 = 2 \times 10^{-4}$. We train network with full batch Gradient descent and set the learning rate to $2 \times 10^{-4}$. The network hidden size is set to 64.

**Figure 6 & Appendix Figure 11.** The networks are trained on inputs depicted in Fig. 5A. Labels are represented as one-hot, i.e. $[1, 0], [1, 0], [0, 1], [0, 1]$. The initialization scale $\sigma^2 = 2 \times 10^{-4}$. We train network with full batch Gradient descent and set the learning rate to $2 \times 10^{-4}$. The network hidden size is set to 64.

**Figure 7.** The networks are trained on inputs depicted in Fig. 6A. Labels are represented as one-hot, i.e. $[1, 0], [1, 0], [0, 1], [0, 1]$. As before the initialization scale $\sigma^2 = 2 \times 10^{-4}$. We train network with full batch Gradient descent and set the learning rate to $2 \times 10^{-4}$. The network hidden size is set to 64.

**Appendix Figure 8.** The networks are trained in CIFAR-10 with the preprocessing outlined in Appendix 3. We use the standard training set and train networks on the standard 10-way classification task with a squared error loss. Networks are trained with a single hidden layer with hidden size set to 1024. The initialization scale $\sigma^2 = 2 \times 10^{-4}$. We train network with (mini) batch SGD with a batch size of 64 and set the learning rate to $2 \times 10^{-4}$.

**Appendix Figure 9.** Networks are trained on the hierarchical task depicted in Fig. 2A and on the XOR problem which is depicted in Fig. 2C. Both networks have biases as indicated in the figure. For the XOR problem labels are represented in a one-hot fashion. I.e. $[1, 0], [1, 0], [0, 1], [0, 1]$. The initialization scale $\sigma^2 = 2 \times 10^{-4}$. We train network with full batch Gradient descent and set the learning rate to $2 \times 10^{-4}$. The network hidden size is set to 64.

## F    APPENDIX: PROOFS AND ADDITIONAL DERIVATIONS

### F.1    PROOF OF PROPOSITION 1 (BIAS MODE WITH STRENGTH $c$)

We work in the one-hot input setting $\mathbf{X} = \mathbf{I}_P$ and define the augmented inputs $\tilde{\mathbf{X}} = [\, c\,\mathbf{1}^T; \mathbf{X}\,] \in \mathbb{R}^{(P+1)\times P}$, where $\mathbf{1}_P \in \mathbb{R}^P$ denote the all-ones vector and $[\cdot; \cdot]$ denotes vertical concatenation. Then

$$\tilde{\boldsymbol{\Sigma}}^x \;=\; \frac{1}{P}\tilde{\mathbf{X}}\tilde{\mathbf{X}}^T \;=\; \frac{1}{P}\begin{bmatrix} (c\mathbf{1}^T)(c\mathbf{1}^T)^T & (c\mathbf{1}^T)\mathbf{X}^T \\ \mathbf{X}(c\mathbf{1}^T)^T & \mathbf{X}\mathbf{X}^T \end{bmatrix} \;=\; \frac{1}{P}\begin{bmatrix} c^2 P & c\,\mathbf{1}^T \\ c\,\mathbf{1} & \mathbf{I}_P \end{bmatrix} \;\in\; \mathbb{R}^{(P+1)\times(P+1)}.$$

We identify its eigenpairs explicitly and define

$$\tilde{\mathbf{v}}_0 \;=\; \begin{bmatrix} 1 \\ \frac{1}{cP}\mathbf{1} \end{bmatrix}.$$

Direct multiplication shows

$$\tilde{\boldsymbol{\Sigma}}^x\,\tilde{\mathbf{v}}_0 \;=\; \frac{1}{P}\begin{bmatrix} c^2 P & c\,\mathbf{1}^T \\ c\,\mathbf{1} & \mathbf{I}_P \end{bmatrix}\begin{bmatrix} 1 \\ \frac{1}{cP}\mathbf{1} \end{bmatrix} \;=\; \frac{1}{P}\begin{bmatrix} c^2 P + \mathbf{1}^T \frac{1}{P}\mathbf{1} \\ c\,\mathbf{1} + \frac{1}{cP}\mathbf{1} \end{bmatrix} \;=\; \begin{bmatrix} c^2 + \frac{1}{P} \\ \frac{c + \frac{1}{cP}}{P}\mathbf{1} \end{bmatrix} \;=\; \Big(c^2 + \tfrac{1}{P}\Big)\tilde{\mathbf{v}}_0.$$

Hence $d_0 = c^2 + \frac{1}{P}$ is an eigenvalue with right eigenvector $\tilde{\mathbf{v}}_0$. Next, for any $\mathbf{q} \in \mathbb{R}^P$ with $\mathbf{1}^T\mathbf{q} = 0$, consider $\tilde{\mathbf{v}} = [\,0; \mathbf{q}\,]$. Then

$$\tilde{\boldsymbol{\Sigma}}^x\tilde{\mathbf{v}} \;=\; \frac{1}{P}\begin{bmatrix} c^2 P & c\,\mathbf{1}^T \\ c\,\mathbf{1} & \mathbf{I}_P \end{bmatrix}\begin{bmatrix} 0 \\ \mathbf{q} \end{bmatrix} \;=\; \frac{1}{P}\begin{bmatrix} c\,\mathbf{1}^T\mathbf{q} \\ \mathbf{q} \end{bmatrix} \;=\; \frac{1}{P}\begin{bmatrix} 0 \\ \mathbf{q} \end{bmatrix} \;=\; \frac{1}{P}\tilde{\mathbf{v}}.$$

Thus the subspace $\{[\,0; \mathbf{q}\,] : \mathbf{1}^T\mathbf{q} = 0\}$ is a $(P-1)$-dimensional eigenspace with eigenvalue $d = \frac{1}{P}$. Finally, the vector $\tilde{\mathbf{v}}_\perp = [\,1; -c\,\mathbf{1}\,]$ satisfies $\tilde{\boldsymbol{\Sigma}}^x\tilde{\mathbf{v}}_\perp = \mathbf{0}$, yielding a simple eigenvalue at $0$. This proves the spectrum

$$\mathrm{spec}\big(\tilde{\boldsymbol{\Sigma}}^x\big) = \Big\{ c^2 + \tfrac{1}{P},\; \underbrace{\tfrac{1}{P}, \ldots, \tfrac{1}{P}}_{P-1},\; 0 \Big\}.$$

We now compute the action of $\tilde{\boldsymbol{\Sigma}}^{yx} = \frac{1}{P}\mathbf{Y}\tilde{\mathbf{X}}^T$ on the normalized top eigenvector $\hat{\mathbf{v}}_0 = \tilde{\mathbf{v}}_0/\|\tilde{\mathbf{v}}_0\|$ with $\|\tilde{\mathbf{v}}_0\|^2 = 1 + \frac{1}{c^2 P}$. Using $\tilde{\mathbf{x}}_i = [\,c; \mathbf{x}_i\,]$,

$$\tilde{\boldsymbol{\Sigma}}^{yx}\,\tilde{\mathbf{v}}_0 \;=\; \frac{1}{P}\sum_{i=1}^{P}\mathbf{y}_i\,\tilde{\mathbf{x}}_i^T\tilde{\mathbf{v}}_0 \;=\; \frac{1}{P}\sum_{i=1}^{P}\mathbf{y}_i\Big(c + \tfrac{1}{cP}\Big) \;=\; \Big(c + \tfrac{1}{cP}\Big)\bar{\mathbf{y}}, \qquad \bar{\mathbf{y}} = \tfrac{1}{P}\sum_{i=1}^{P}\mathbf{y}_i.$$

Therefore

$$\Big\|\tilde{\boldsymbol{\Sigma}}^{yx}\,\hat{\mathbf{v}}_0\Big\| \;=\; \frac{c + \frac{1}{cP}}{\sqrt{1 + \frac{1}{c^2 P}}}\,\|\bar{\mathbf{y}}\| \;=\; \sqrt{c^2 + \tfrac{1}{P}}\,\|\bar{\mathbf{y}}\| \;=:\; s_0,$$

which is the top singular value associated with the bias direction. This establishes equation 5 and completes the proof.

### F.2    ADDITIONAL NOTES ON RAPID LINEARIZATION

The final approximate gradient updates to the bias and first layer weights at initialisation are:

$$\dot{\mathbf{W}}_0^1 \approx \epsilon^2\Big(\mathbf{H}\mathbf{Y}^T\mathbf{Y} \odot \sigma'(\mathbf{Z})\Big)\mathbf{X}^T; \qquad \dot{\mathbf{b}}_0 \approx \epsilon^2\Big(\mathbf{H}\mathbf{Y}^T\mathbf{Y} \odot \sigma'(\mathbf{Z})\Big)\mathbf{1} \tag{24}$$

While it is sufficient in Section 5.3 to consider the case of $\mathbf{Y}$ being element-wise positive semi-definite, as this is by far the most common form of most datasets, it can be constructive to consider the case of having some negative elements in $\mathbf{Y}$ here. If there are negative output labels then $\mathbf{Y}^T\mathbf{Y}$ is the matrix of dot products between all outputs, providing a similarity measure between the outputs. Thus, the elements of $\mathbf{Y}^T\mathbf{Y}\mathbf{1}$ provides a measure for each data point on how similar its output is to all other outputs vectors. If it is similar then the value will be positive. The update to the $i$-th bias parameter then will be $\mathbf{h}_{[:,i]}\mathbf{Y}^T\mathbf{Y}\mathbf{1}$ and will be positive when the neuron is active for data points with similar outputs. Importantly, this gradient does not depend on the bias alone but will occur for

all active neurons in the hidden layer. This explains why many units with slightly negative initial $b_j$ still drift upward (slower), consistent with our experiments. This upward drift of the bias parameters then makes it more likely for the neurons to be active in subsequent epochs of training, resulting in a more linear network emerging in the initial stage of training. This, at least partially, explains why we do not see the time shift between the ReLU and linear network dynamics implied by the results of Section 5.4. Even if many neurons begin by being gated off for many data points, the early learning bias from small initial weights will cause the biases to increase which leads to more active neurons for each data point and a closer match between the linear and ReLU network learning dynamics.

## F.3 DERIVATION OF HITTING TIME

We will begin by using the definition of hitting time and substituting the mode dynamics (Equation 7) into this expression to obtain an expression for $t$:

$$a_\alpha = \rho$$

$$\frac{s_\alpha/d_\alpha}{1 - \left(1 - \frac{s_\alpha}{d_\alpha a_0}\right)\exp\left(\frac{-2s_\alpha}{\tau}t\right)} = \rho$$

$$\frac{s_\alpha/d_\alpha}{\rho} = 1 - \left(1 - \frac{s_\alpha}{d_\alpha a_0}\right)\exp\left(\frac{-2s_\alpha}{\tau}t\right)$$

$$\frac{\rho - s_\alpha/d_\alpha}{\rho} = \left(1 - \frac{s_\alpha}{d_\alpha a_0}\right)\exp\left(\frac{-2s_\alpha}{\tau}t\right)$$

$$\frac{\rho - s_\alpha/d_\alpha}{\rho\left(1 - \frac{s_\alpha}{d_\alpha a_0}\right)} = \exp\left(\frac{-2s_\alpha}{\tau}t\right)$$

$$\log\left(\frac{\rho - s_\alpha/d_\alpha}{\rho\left(1 - \frac{s_\alpha}{d_\alpha a_0}\right)}\right) = \frac{-2s_\alpha}{\tau}t$$

$$\frac{-\tau}{2s_\alpha}\log\left(\frac{\rho - s_\alpha/d_\alpha}{\rho\left(1 - \frac{s_\alpha}{d_\alpha a_0}\right)}\right) = t$$

$$\frac{-\tau}{2s_\alpha}\log\left(\frac{\rho\left(1 - \frac{s_\alpha}{d_\alpha\rho}\right)}{\rho\left(1 - \frac{s_\alpha}{d_\alpha a_0}\right)}\right) = t$$

$$\frac{\tau}{2s_\alpha}\log\left(\frac{1 - \frac{s_\alpha}{d_\alpha a_0}}{1 - \frac{s_\alpha}{d_\alpha\rho}}\right) = t$$

$$\frac{\tau}{2s_\alpha}\log\left(\frac{\frac{s_\alpha}{d_\alpha a_0} - 1}{\frac{s_\alpha}{d_\alpha\rho} - 1}\right) = t$$

$$\frac{\tau}{2s_\alpha}\log\left(\frac{\frac{s_\alpha - d_\alpha a_0}{d_\alpha a_0}}{\frac{s_\alpha - d_\alpha\rho}{d_\alpha\rho}}\right) = t$$

$$\frac{\tau}{2s_\alpha}\log\left(\frac{s_\alpha d_\alpha\rho - d_\alpha^2 a_0\rho}{s_\alpha d_\alpha a_0 - d_\alpha^2 a_0\rho}\right) = t$$

$$\frac{\tau}{2s_\alpha}\log\left(\frac{s_\alpha\rho - d_\alpha a_0\rho}{s_\alpha a_0 - d_\alpha a_0\rho}\right) = t$$

The term inside of the log is always going to be greater than 1 as $\rho \geq a_0$. Thus the log is positive. In the extreme case where $\rho = a_0$ then the log evaluates to 0 as the internal fraction is 1, which makes sense as in this case the hitting time will be reached at initialization.

F.4  HITTING TIME DELAY FROM GATING

The derivation in Appendix F.3 shows that the hitting time for a mode depends on the initial singular value for the mode ($a_0$). Importantly, this is the only variables which depends on the network architecture or parameters for the hitting time. All other variables are determined by the dataset statistics. To begin we define two network mappings:

$$\hat{Y}_{linear} = \mathbf{W}_2\mathbf{W}_1 = \mathbf{U}\sqrt{\mathbf{S}}\mathbf{R}^T\mathbf{R}\sqrt{\mathbf{S}}\mathbf{V}^T; \qquad \hat{Y}_{relu} = \mathbf{W}_2\mathbf{W}_1 = \mathbf{U}\sqrt{\mathbf{S}}\tilde{\mathbf{R}}^T\mathbf{G}\tilde{\mathbf{R}}\sqrt{\mathbf{S}}\mathbf{V}^T. \quad (25)$$

We are assuming that both networks align to the singular vectors of the dataset correlation matrices and that the ReLU mapping is able to learn a linear pathway at initialisation (which is guaranteed by the presence of the bias parameters). The gating matrix $\mathbf{G}$ gates off neurons that are dead at initialisation providing an approximation for the behaviour of a ReLU network. We can compute the initial value for the $i$-th effective singular value as:

$$\mathbf{W}_2\mathbf{W}_1 = \mathbf{U}\sqrt{\mathbf{A}}\mathbf{R}^T\mathbf{R}\sqrt{\mathbf{A}}\mathbf{V}^T$$

$$\mathbf{U}_i^T\mathbf{W}_2\mathbf{W}_1\mathbf{V}_i = \mathbf{U}_i^T\mathbf{U}\sqrt{\mathbf{A}}\mathbf{R}^T\mathbf{R}\sqrt{\mathbf{A}}\mathbf{V}^T\mathbf{V}_i$$

$$= \sqrt{a_0}\mathbf{R}_i^T\mathbf{R}_i\sqrt{a_0} = a_0$$

Above we used the fact that $\mathbf{R}_i^T\mathbf{R}_i = 1$ Similarly for the ReLU mappings:

$$\mathbf{W}_2\mathbf{G}\mathbf{W}_1 = \mathbf{U}\sqrt{\mathbf{A}}\tilde{\mathbf{R}}^T\mathbf{G}\tilde{\mathbf{R}}\sqrt{\mathbf{A}}\mathbf{V}^T$$

$$\mathbf{U}_i^T\mathbf{W}_2\mathbf{G}\mathbf{W}_1\mathbf{V}_i = \mathbf{U}_i^T\mathbf{U}\sqrt{\mathbf{A}}\tilde{\mathbf{R}}^T\mathbf{G}\tilde{\mathbf{R}}\sqrt{\mathbf{A}}\mathbf{V}^T\mathbf{V}_i$$

$$= \sqrt{a_0}\tilde{\mathbf{R}}_i^T\mathbf{G}\tilde{\mathbf{R}}_i\sqrt{a_0} = a_0\sum_{j=1}^{N}g_j r_j^2$$

Since we sample initial weights from an isotropic gaussian all values of $r_j$ are equal in expectations. Thus let $r_j = \bar{r}$ and let $\bar{g}$ be the dead neuron rate at initialisation:

$$\mathbf{U}_i^T\mathbf{W}_2\mathbf{G}\mathbf{W}_1\mathbf{V}_i = a_0\sum_{j=1}^{N}g_j r_j^2 = a_0\sum_{j=1}^{N}\bar{g}\bar{r}^2 = a_0 N\bar{g}\bar{r}^2 = a_0\bar{g}$$

Since:

$$\sum_{j=1}^{N}r_j^2 = \sum_{j=1}^{N}\bar{r}^2 = N\bar{r}^2 := 1$$

Substituting these two initial values into the hitting time equations for the linear and ReLU networks we obtain the hitting time for each:

$$t_{linear} = \frac{\tau}{2s_\alpha}\log\left(\frac{s_\alpha\rho - d_\alpha a_0\rho}{s_\alpha a_0 - d_\alpha a_0\rho}\right) \quad (26)$$

and

$$t_{relu} = \frac{\tau}{2s_\alpha}\log\left(\frac{s_\alpha\rho - d_\alpha a_0\bar{g}\rho}{s_\alpha a_0\bar{g} - d_\alpha a_0\bar{g}\rho}\right) \quad (27)$$

Thus the difference in hitting time from gating is:

$$t_{relu} - t_{linear} = \frac{\tau}{2s_\alpha}\log\left(\frac{s_\alpha\rho - d_\alpha a_0\bar{g}\rho}{s_\alpha a_0\bar{g} - d_\alpha a_0\bar{g}\rho}\right) - \frac{\tau}{2s_\alpha}\log\left(\frac{s_\alpha\rho - d_\alpha a_0\rho}{s_\alpha a_0 - d_\alpha a_0\rho}\right)$$

$$= \frac{\tau}{2s_\alpha}\left[\log\left(\frac{s_\alpha\rho - d_\alpha a_0\bar{g}\rho}{s_\alpha a_0\bar{g} - d_\alpha a_0\bar{g}\rho}\right) - \log\left(\frac{s_\alpha\rho - d_\alpha a_0\rho}{s_\alpha a_0 - d_\alpha a_0\rho}\right)\right]$$

$$= \frac{\tau}{2s_\alpha}\log\left(\left(\frac{s_\alpha\rho - d_\alpha a_0\bar{g}\rho}{s_\alpha a_0\bar{g} - d_\alpha a_0\bar{g}\rho}\right)\left(\frac{s_\alpha a_0 - d_\alpha a_0\rho}{s_\alpha\rho - d_\alpha a_0\rho}\right)\right)$$

$$= \frac{\tau}{2s_\alpha}\log\left(\frac{s_\alpha^2 a_0 - s_\alpha d_\alpha a_0^2\bar{g} - s_\alpha d_\alpha a_0\rho + d_\alpha^2 a_0^2\bar{g}\rho}{s_\alpha^2 a_0\bar{g} - s_\alpha d_\alpha a_0\bar{g}\rho - s_\alpha d_\alpha a_0^2\bar{g} + d_\alpha^2 a_0^2\bar{g}\rho}\right)$$

Finally we can prove that this change in hitting time will always be positive by considering the terms inside of the logarithm:

$$\frac{s_\alpha^2 a_0 - s_\alpha d_\alpha a_0^2 \bar{g} - s_\alpha d_\alpha a_0 \rho + d_\alpha^2 a_0^2 \bar{g} \rho}{s_\alpha^2 a_0 \bar{g} - s_\alpha d_\alpha a_0 \bar{g} \rho - s_\alpha d_\alpha a_0^2 \bar{g} + d_\alpha^2 a_0^2 \bar{g} \rho} > 1$$

$$s_\alpha^2 a_0 - s_\alpha d_\alpha a_0^2 \bar{g} - s_\alpha d_\alpha a_0 \rho + d_\alpha^2 a_0^2 \bar{g} \rho > s_\alpha^2 a_0 \bar{g} - s_\alpha d_\alpha a_0 \bar{g} \rho - s_\alpha d_\alpha a_0^2 \bar{g} + d_\alpha^2 a_0^2 \bar{g} \rho$$

$$s_\alpha^2 a_0 (1 - \bar{g}) - s_\alpha d_\alpha a_0 \bar{g} (a_0 - \rho) - s_\alpha d_\alpha a_0 (\rho - a_0 \bar{g}) > 0$$

$$s_\alpha (1 - \bar{g}) - d_\alpha \bar{g} (a_0 - \rho) - d_\alpha (\rho - a_0 \bar{g}) > 0$$

$$s_\alpha (1 - \bar{g}) - d_\alpha (a_0 \bar{g} - \bar{g} \rho + \rho - a_0 \bar{g}) > 0$$

$$s_\alpha (1 - \bar{g}) - d_\alpha \rho (1 - \bar{g}) > 0$$

$$s_\alpha > d_\alpha \rho$$

Since $\rho$ is a small controllable parameter it is always possible to satisfy this condition. Thus, all terms in the change of hitting time equation will be positive and dead neurons will always slow down learning.

