# OpenReview forum: "Bias-mediated early linearization drives simplicity bias in ReLU networks"
_ICLR.cc/2026/Conference — ICLR 2026 Conference Withdrawn Submission_

### Official Review · Reviewer_H5cq · 2025-10-27

**Soundness:** 1
**Presentation:** 1
**Contribution:** 1
**Rating:** 2
**Confidence:** 4

**Summary:**

The paper focuses on the effect of the existence of bias terms in the ReLU neural networks. It argues that, in the small initialization setting, the bias term dominates the training dynamics in the early stage and induces a similarity between the dynamics of ReLU network and linear network.

**Strengths:**

Looking at the difference made by the bias terms in the dynamics is a interesting perspective. I also noticed that there are some prior work [1] which suggesting the importance of the bias terms.

The paper is easy to follow.


[1]: Freeze and Chaos: NTK views on DNN Normalization, Checkerboard and Boundary Artifacts, Jacot, Gabriel, Ged, Hongler

**Weaknesses:**

1: The paper fails to substantiate one of its central claims—namely, that “ReLU and linear networks, when equipped with bias terms, exhibit equivalent (early) learning dynamics.” This conclusion is drawn primarily from visual comparisons of training loss curves (e.g., Figure 1). However, the similarity of loss curves between two models does not necessarily imply that their learning dynamics are equivalent. Since loss curves typically decrease over training time, it is common for different models or algorithms to display similar training curves despite having fundamentally distinct underlying dynamics. To convincingly support the claim of equivalent learning dynamics, the paper should analyze additional metrics that directly reflect the training process—such as the similarity between the parameter trajectories ${w_t}$.

2: The paper compares the ReLU network with a linear network and repeatedly asserts that the latter is a linear model with well-understood linear dynamics. This is not correct. It has been known for a while [2] that linear networks have **nonlinear** dynamics, and,  based on the belief that the nonlinear dynamics of the linear network shares a lot of similarity with that of nonlinear networks, they are often used as prototypes of nonlinear deep networks for studying their nonlinear learning dynamics [3, and reference therein]. Note that the dynamics of both linear and nonlinear networks are **not** "well-understood", although the former might be a bit mathematically easier to analyze. The author should also be aware that these dynamics are far from the dynamics of a linear model.
The paper should distinguish the dynamics of a linear model and that of a linear network, to avoid confusion.

3: The experiments and analysis are limited to the small-initialization setting, under which the results of the paper become rather straightforward. In small-initialization setting, $Wx$ becomes small $o(1)$ ($W$ is the layer weight matrix, $x$ is the layer's input -- e.g., previous hidden layer). Therefore, in the computation of next layer neuron $Wx + b\approx b$ ($b$ is the bias). Basically, that means the contribution of the weights is minimal (at least at the beginning of training, before $Wx$ getting big). Similar bias dominance also holds in back-propagation. Consequently, it is unsurprising that the bias terms dominate the learning dynamics until the weights grow sufficiently large.

4: The theoretical analysis is limited to one-hot inputs. It is unclear whether the theory generalizes to practical settings.

[2]: Exact solutions to the nonlinear dynamics of learning in deep linear neural networks, Saxes, McClelland, Ganguli

[3]: The Law of Parsimony in Gradient Descent for Learning Deep Linear Networks, Yaras, Wang, Hu, Zhu, Balzano, Qu

**Questions:**

Q1: At Line 314, it promises to understand the alignment to $\bar{y}$. However, I could not locate the result and discussion regarding this promise.

Q2: Can you discuss the similarity/difference with the paper: When Are Bias-Free ReLU Networks Effectively Linear Networks? Zhang, Saxe, Latham.

---

### Official Review · Reviewer_L5i2 · 2025-11-01

**Soundness:** 2
**Presentation:** 2
**Contribution:** 1
**Rating:** 2
**Confidence:** 4

**Summary:**

This paper investigates the simplicity bias of ReLU networks, with a specific focus on the role of the bias term. The authors demonstrate that ReLU networks equipped with a bias term find simpler solutions than their counterparts without bias. A key finding is that in the early phase of training, the dynamics of a ReLU network with bias align with those of a linear network.

**Strengths:**

* This work presents an interesting and novel relationship between the bias term and the simplicity bias of ReLU networks.
* This relationship is well-supported by (1) a theoretical analysis in a simple setting (one-hot inputs) and (2) empirical validation on both synthetic and real data.
* The experiments on the 3D XOR dataset were particularly effective in providing clear, intuitive visual support for the paper's findings.

**Weaknesses:**

My primary concern is that while the paper's insights are interesting, its scope is too limited.
* In Section 5.3, the explanations for why the bias term leads to linearization are based on approximation and intuition, rather than on concrete theoretical guarantees.
* The results in Section 5.4 (regarding hitting time) heavily rely on adapting Equation (7) which is proven in previous work (Saxe et al., 2014; 2019). It appears the authors assume the rate of dead neurons $\bar g$ is constant over time, which allows them to simply apply results from linear networks. This is a strong simplifying assumption that limits the novelty.
* I acknowledge that developing rigorous theory for deep ReLU networks is challenging. However, given this theoretical limitation (even for two-layer networks), the paper should have compensated with more extensive empirical investigations beyond two-layer networks to demonstrate the generality of the findings.

Due to combination of an absence of rigorous theory (even for the two-layer ReLU case) and limited experimental settings, I believe that the overall contribution may not be significant enough to meet the acceptance bar for ICLR.

### Minors
* In Equation (2), $\tilde W^1$->$W^1$
* In line 304, ''silent alignment effect'' -> ``silent alignment effect''
* In line 1034,  matrix $P$ -> $I_P$ in the left upper block matrix.

**Questions:**

I have a question regarding the proof of Proposition 1, specifically the characterization of the first singular value of $\tilde \Sigma^{yx}$. The proof seems to use the fact that $\tilde v_0$, which is defined as the top singular vector of $\tilde \Sigma^x$, is also treated as the top singular vector of $\tilde \Sigma^{yx}$. Could the author clarify why this holds? I think this step is not obvious.

---

### Official Review · Reviewer_jDtk · 2025-11-02

**Soundness:** 3
**Presentation:** 4
**Contribution:** 3
**Rating:** 8
**Confidence:** 4

**Summary:**

The paper investigates simplicity biases in neural networks, specifically focusing on the role of the bias term in ReLU networks. It claims that the bias term drives the early learning dynamics into a phase that mimics the training of fully linear networks. The authors theoretically demonstrate that during this initial phase, the bias terms are primarily learned, effectively pushing inputs into the linear region of the ReLU activation. This phenomenon makes the overall training dynamics resemble those of a linear model. Finally, the paper shows that ReLU networks with bias terms naturally converge towards linear solutions if such solutions exist, in contrast to networks without bias terms, which tend to converge towards non-linear solutions.

**Strengths:**

- **Originality:** The connection between simplicity biases in neural networks and the role of bias terms is a novel and highly interesting contribution. While inductive simplicity biases has been explored in various works, this specific link appears to be an original perspective.
- **Clarity:** The manuscript is well-written and easy to follow. The figures are also very clear and aid understanding the results and setups. Overall, it was a pleasure to read.
- **Quality:** The study spans theoretical analysis of training dynamics, then thoroughly validated empirically using both real and natural datasets (MNIST and gray-scale CIFAR10).
- **Significance:** The characterization of the training dynamics of ReLU networks into an initial phase mimicking linear behavior is a valuable insight that can aid future theoretical analyses of neural networks.

**Weaknesses:**

- **Significance:** As acknowledged by the authors in the conclusion, the presented analysis is currently limited to relatively small-scale two-layer networks and datasets (MNIST and gray-scale CIFAR10). While valuable, scaling these insights to larger, more complex architectures and datasets would strengthen the generalizability of the findings.
- **Significance:** The current study is exclusively focused on ReLU activations. It would be beneficial if the authors could comment on the potential extension of their insights to other ReLU-like activation functions, such as GeLU, which is increasingly prevalent in modern architectures like Transformers. If the core insight is that the presence of bias terms drive towards linear behavior, similar effects might be observed with GeLU, and a discussion on this would enrich the paper.

**Questions:**

1. Recent papers have explored the simplicity bias of ReLU networks from different perspectives on the components that play a role. For instance, works like [A] and [B] have highlighted the effect of weight initialization (e.g., centered around 0, which is also considered in the setup of the paper) in influencing network behavior. Do you think that the linear phase could also be linked to the initialization of the weights? Would we observe similar behaviors for other activations like Sigmoid or tanh that has their non-linear gating mechanisms outside 0 with proper initialization shifted towards the non-linear inflection points?
2. Do you anticipate similar observations regarding the linear behavior for other ReLU-like activation functions?
3. To what extent do you believe these results generalize when scaling up to deeper networks with more layers? Do you think that having more activations functions throughout the network (more than one) can impact the learning dynamics?

### References

- [A] Teney et al. "Neural redshift: Random networks are not random functions." CVPR 2024.
- [B] Bouniot et al. "From alexnet to transformers: Measuring the non-linearity of deep neural networks with affine optimal transport." CVPR 2025.

---

### Official Review · Reviewer_btQ9 · 2025-11-04

**Soundness:** 3
**Presentation:** 2
**Contribution:** 2
**Rating:** 2
**Confidence:** 3

**Summary:**

The paper argues that the use of bias terms makes ReLU networks' behave like linear networks early in training. The explanation is roughly as follows: if we start with small initialization the initial output is near 0. If the labels are positive, the weights in a two layer ReLU network will have a tendency to increase (rather than decrease) under gradient flow/descent because the output of the network should be increased and the output of the hidden layer is non-negative. In particular, the bias terms will increase (accept those for neurons that are not active on any input). Once the bias terms get larger, they will contribute significantly and positively to the inputs of the ReLU making the ReLU operate in its linear region.

**Strengths:**

Beyond making the point that initial training is nearly linear, the authors also make the case that this improves simplicity bias.

**Weaknesses:**

The general setup seems slightly unusual. For example using MNIST with one-hot encoded labels and mean square loss. In a way, it is very much designed to obtain the desired effect. Larger more realistic experiments are lacking.

The writing could be improved. There are a number of typos. Some of them change the intended meaning in some fundamental way (e.g. in line 411 it says "with bias" when I assume it should say "without bias"). Figure 6 is difficult to read.

The main paper refers to figures (e.g. to Figure 10) in the appendix and therefore cannot really be read independently.

**Questions:**

Can you produce experiments in more realistic settings (bigger, deeper networks and bigger datasets with more typical loss functions for the corresponding task) that show the same effects?

---

### Note · Authors · 2025-11-24

**Comment:**

Dear Reviewers,

Thank you very much for the time and care you devoted to reviewing our submission. We truly appreciate the constructive and insightful feedback you provided. Your comments highlighted several important areas where our work can be strengthened, and they have given us a clear path toward improving both the clarity and rigor of the manuscript.

After considering the reviews, we have decided to withdraw the paper from the current ICLR review cycle. We believe this will allow us to thoroughly address the concerns raised, incorporate your suggestions, and further develop the ideas in a more robust and compelling way. We are grateful for your efforts and for the valuable guidance you have offered. Thank you again for helping us improve our work.

Best,

the authors

**Withdrawal Confirmation:**

I have read and agree with the venue's withdrawal policy on behalf of myself and my co-authors.